# MULTI-AGENT DEEP REINFORCEMENT LEARNING WITH EXTREMELY NOISY OBSERVATIONS

## ABSTRACT

Multi-agent reinforcement learning systems aim to provide interacting agents with the ability to collaboratively learn and adapt to the behaviour of other agents. In many real-world applications, the agents can only acquire a partial view of the world. Here we consider a setting whereby most agents' observations are also extremely noisy, hence only weakly correlated to the true state of the environment. Under these circumstances, learning an optimal policy becomes particularly challenging, even in the unrealistic case that an agent's policy can be made conditional upon all other agents' observations. To overcome these difficulties, we propose a multi-agent deep deterministic policy gradient algorithm enhanced by a communication medium (MADDPG-M), which implements a two-level, concurrent learning mechanism. An agent's policy depends on its own private observations as well as those explicitly shared by others through a communication medium. At any given point in time, an agent must decide whether its private observations are sufficiently informative to be shared with others. However, our environments provide no explicit feedback informing an agent whether a communication action is beneficial, rather the communication policies must also be learned through experience concurrently to the main policies. Our experimental results demonstrate that the algorithm performs well in six highly non-stationary environments of progressively higher complexity, and offers substantial performance gains compared to the baselines.

## 1 INTRODUCTION

Reinforcement Learning (RL) is concerned with enabling agents to learn how to accomplish a task by taking sequential actions in a given stochastic environment so as to maximise some notion of cumulative reward, and relies on Markov Decision Processes (MDP) (Sutton et al., 1998). The decision maker, or agent, follows a policy defining which actions should be chosen under each environmental state. In recent years, Deep Reinforcement Learning (DRL), which leverages RL approaches using Deep Neural Network based function approximators, has been proved to achieve human-level performance in a number of applications, mostly gaming environments, requiring an individual agent (Mnih et al., 2015; Silver et al., 2016). Many real-world applications, on the other hand, can be modelled as multi-agent systems, e.g. autonomous driving (Dresner & Stone, 2008), energy management (Jun, 2011; Col, 2011), fleet control (Stranjak et al., 2008), trajectory planning (Bento et al., 2013), and network packet routing (Di Caro et al., 1998). In such cases, the agents interact with each other to successfully accomplish the underlying task. Straightforward applications of single-agent DRL methodologies for learning a multi-agent policy are not well suited for a number of reasons. Firstly, from the point of view of an individual agent, the environment behaves in a highly non-stationary manner as it now depends not only on the agent's own actions, but also on the joint action of all other agents (Lowe et al., 2017). The severe non-stationarity violates the Markov assumption and prevents the naive use of experience replay (Lin, 1992), which is important to stabilise training and improve sample efficiency. Furthermore, only rarely each individual agent has a complete and accurate representation of the entire environment. Typically, an agent receives its own private observations providing only a partial view of the true state of the world. Determining which agent should be credited for the observed reward is also non-trivial (Foerster et al., 2018).

Training each agent independently, thus effectively ignoring the non-stationarity, is the simplest possible approach. Independent Q-Learning (IQL) (Tan, 1993) leverages traditional Q-learning in this fashion, and some empirical success has been reported (Matignon et al., 2012) despite convergence

issues. Tampuu et al. (2017) has extended IQL for Deep Q-learning to play Pong in a competitive multi-agent setting, and Tesauro (2003) has addressed the non-stationarity problem by allowing each agent's policy to depend upon the estimates of the other agents' policies. The *Centralised Training Decentralised Execution* (CTDE) paradigm has been widely adopted to overcome the non-stationarity problem when training multi-agent systems (Foerster et al., 2016). CTDE enables to leverage the observations of each agent, as well as their actions, to better estimate the action-value functions during training. As the policy of each agent only depends on its own private observations during training, the agents are able to decide which actions to take in a decentralised manner during execution. Recently, Lowe et al. (2017) have combined this paradigm with a Deep Deterministic Policy Gradient (DDPG) algorithm (Lillicrap et al., 2015) to solve Multi-agent Particle Environments, and Foerster et al. (2018) have used a similar approach integrated with a counter-factual baseline to address the credit-assignment problem.

In this article we consider a particularly challenging multi-agent learning problem whereby all agents operate under partial information, and the observations they receive are weakly correlated to the true state of the environment. Similar settings arise in real-world applications, e.g. in cooperative and autonomous driving, when an agent's view of the world at any given time, obtained through a number of sensors, may carry a high degree of uncertainty and can occasionally be wrong. Learning to collaboratively solve the underlying task under these conditions becomes unfeasible unless an appropriate information filtering mechanism is in place allowing only the accurate observations to be shared across agents and inform their policies. The rationale is that, when an observation is accurate, sharing it with others will progressively contribute to form a better view of the world and make more educated decisions, whereas indiscriminately sharing of all the information can be detrimental due to the high level of noise. To keep the setting realistic, we do not assume that the agents are explicitly told whether their private observations are accurate or noisy, rather they need to discover this through experience.

We propose a multi-agent deep deterministic policy gradient algorithm enhanced by a communication medium (MADDPG-M) to address these requirements. During training, each agent's policy depends on its own observations as well as those explicitly shared by other agents; every agent simultaneously learns whether its current private observations contribute to maximising future expected rewards, and therefore are worth sharing with others at an given time, whilst also collaboratively learning the underlying task. Extensive experimental results demonstrate that MADDPG-M performs well in highly non-stationary environments, even when the agents acquiring relevant observations continuously change within an episode. As the execution operates in a decentralised manner, the algorithm complexity per time-step is linear in the number of agents. In order to assess the performance of MADDPG-M, we have designed and tested a number of environments as extensions of the original *Cooperative Navigation* problem (Lowe et al., 2017). Each agent is a 2D object aiming to reach a different landmark while avoiding collisions with other agents. We consider two types of settings. In the first one, a single "gifted" agent can see the true location of all the landmarks, whilst the other agents receive their wrong whereabouts. In the second one, the agents may have information that is only valuable to other agents, and has to be redirected to the right recipient. We compare MADDPG-M against existing baselines on all the environments and discuss its relative merits and potential future improvements.

## 2 RELATED WORK

Multi-agent Deep Deterministic Policy Gradient (MADDPG) (Lowe et al., 2017) adopts the CTDE paradigm and builds a multi-agent approach upon DDPG (Lillicrap et al., 2015) to solve mixed cooperative-competitive environments. No explicit communication is allowed. Instead, centralised training helps agents learn a coordinated behaviour. The CTDE paradigm has been shown to work well on Multi-agent Particle Environments (Lowe et al., 2017) and *StarCraft unit micromanagement* (Foerster et al., 2018). Extensive efforts have also been spent towards enabling communication amongst agents. The large majority of existing methods enable communication by introducing a differential channel through which the gradients can be sent across agents. For instance, in Differentiable Inter-agent Learning (DIAL) (Foerster et al., 2016), the agents communicate in a discrete manner through their actions. DIAL uses Q-learning based Deep Recurrent Q-Networks (DRQN) (Hausknecht & Stone, 2015) with weight sharing. The algorithm is end-to-end trainable across agents due to its ability to pass gradients from agent to agent over the messages, which

requires a communication medium scaling quadratically with the number of agents. This approach has been used to solve problems such as switch riddle where communicating over 1-bit messages is sufficient. Analogously to DIAL, Communication Neural Net (CommNet) (Sukhbaatar et al., 2016) uses a parameter-sharing strategy across agents and defines a differentiable communication channel between them. Every agent has access to this shared channel carrying the average of the messages of all agents. CommNet uses a large single network for all the agents, which may not be easily scalable. Multi-agent Bidirectionally-Coordinated Network (BiCNet) (Peng et al., 2017) adopts a recurrent neural network-based memory to form a communication channel among agents and uses a centralised control policy conditioning on the true state. Other authors have also studied the emergence of language in multi-agent systems (Lazaridou et al., 2016; Mordatch & Abbeel, 2018; Lazaridou et al., 2018). In these works, the environment typically provides explicit feedback about the communication actions. Unlike existing studies, we do not assume the existence of explicit rewards guiding the communication actions. Our problem therefore requires the hierarchical arrangement of communication policies and local agent policies that act on the environment, which must be learned concurrently. Furthermore, we consider settings where the observations are either wrong or randomly allocated across agents so that, without an appropriate communication strategy, no optimal policies can be learned.

## 3 BACKGROUND

### 3.1 PARTIALLY OBSERVABLE MARKOV GAMES

Partially observable Markov Games (POMGs) (Littman, 1994) are multi-agent extensions of MDPs consisting of $N$ agents with partial observations and characterised by a set of true states $\mathcal{S}$, a collection of action sets $\mathcal{A} = \{\mathcal{A}_1, \ldots, \mathcal{A}_N\}$, a state transition function $\mathcal{T}$, a reward function $\mathcal{R}$, a collection of private observation functions $\mathcal{Q} = \{\mathcal{Q}_1, \ldots, \mathcal{Q}_N\}$, a collection of private observations $\mathcal{O} = \{\mathcal{O}_1, \ldots, \mathcal{O}_N\}$ and a discount factor $\gamma \in [0, 1)$. A POMG is then defined by a tuple, $G = \langle \mathcal{S}, \mathcal{A}, \mathcal{T}, \mathcal{R}, \mathcal{Q}, \mathcal{O}, \gamma, N \rangle$. The agents do not have full access to the true state of the environment $s \in \mathcal{S}$, instead each receives a private partial observation correlated with the true state, i.e. $o_i = \mathcal{Q}_i(s) : \mathcal{S} \to \mathcal{O}_i$. Each agent chooses an action according to a (either deterministic or stochastic) policy parameterised by $\theta_i$ and conditioned on its own private observation, i.e. $a_i = \boldsymbol{\mu}_{\theta_i}(o_i) : \mathcal{O}_i \to \mathcal{A}_i$ or $\boldsymbol{\pi}_{\theta_i}(a_i|o_i) : \mathcal{O}_i \times \mathcal{A}_i \to [0, 1]$, and obtains a reward as a result of this action, i.e. $r_i = \mathcal{R}(s, a_i) : \mathcal{S} \times \mathcal{A}_i \to \mathbb{R}$. The environment then moves into the next state $s' \in \mathcal{S}$ according to the state transition function conditioned on actions of all agents, i.e. $s' = \mathcal{T}(s, a_1, \ldots, a_N) : \mathcal{S} \times \mathcal{A}_1 \times \ldots \times \mathcal{A}_N \to \mathcal{S}$. Each agent aims to maximise its own total expected return, $\mathbb{E}[R_i] = \mathbb{E}[\sum_{t=0}^{T} \gamma^t r_i^t]$ where $r_i^t$ is the collected reward by the $i^{\text{th}}$ agent at time $t$ and $T$ is the time horizon.

### 3.2 DETERMINISTIC POLICY GRADIENT ALGORITHMS

Policy Gradient (PG) algorithms are based on the idea that updating the policy's parameter vector $\theta$ in the direction of $\nabla_\theta J(\theta)$ maximises the objective function, $J(\theta) = \mathbb{E}[R]$. The current policy $\boldsymbol{\pi}_\theta$ is specified by a stochastic function using a set of probability measures on the action space, i.e. $\boldsymbol{\pi}_\theta : \mathcal{S} \to \mathcal{P}(\mathcal{A})$. Deterministic Policy Gradient (DPG) (Silver et al., 2014) extends the policy gradient framework by adopting a policy function that deterministically maps states to actions, i.e. $\boldsymbol{\mu}_\theta : \mathcal{S} \to \mathcal{A}$. The gradient used to optimise the objective $J(\theta)$ is

$$\nabla_\theta J(\theta) = \mathbb{E}_{s \sim \mathcal{D}}[\nabla_\theta \boldsymbol{\mu}_\theta(a|s) \nabla_a Q^{\boldsymbol{\mu}}(s, a)|_{a=\boldsymbol{\mu}_\theta(s)}] \tag{1}$$

where $\mathcal{D}$ is an experience replay buffer and $Q^{\boldsymbol{\mu}}(s, a)$ is the corresponding action-value function. In DPG, the policy gradient takes the expectation only over the state space, which introduces data efficiency advantages. On the other hand, as the policy gradient also relies on $\nabla_a Q^{\boldsymbol{\mu}}(s, a)$, DPG requires a continuous policy $\boldsymbol{\mu}$. Deep Deterministic Policy Gradient (DDPG) (Lillicrap et al., 2015) builds upon DPG by adopting Deep Neural Networks to approximate $\boldsymbol{\mu}$ and $Q^{\boldsymbol{\mu}}$. Analogously to DQN (Mnih et al., 2015), the experience replay buffer $\mathcal{D}$ and target networks $\boldsymbol{\mu}', Q^{\boldsymbol{\mu}'}$ help stabilise the learning.

### 3.3 MULTI-AGENT DEEP DETERMINISTIC POLICY GRADIENT

Multi-agent Deep Deterministic Policy Gradient (MADDPG) extends DDPG to multi-agent settings by adopting the CTDE paradigm. DDPG is well-suited for such an extension as both $\boldsymbol{\mu}$ and $Q^{\boldsymbol{\mu}}$ can be made dependent upon external information. In order to prevent non-stationarity, MADDPG uses the actions and observations of all agents in the action-value functions, $Q_i^{\boldsymbol{\mu}}(o_1, a_1, \ldots, o_N, a_N)$. On the other hand, as the policy of an agent is only conditioned upon its own private observations, $a_i = \boldsymbol{\mu}_{\theta_i}(o_i)$, the agents can act in a decentralised manner during execution. The gradient of the continuous policy $\boldsymbol{\mu}_{\theta_i}$ (hereafter abbreviated as $\boldsymbol{\mu}_i$) with respect to parameters $\theta_i$ is

$$\nabla_{\theta_i} J(\boldsymbol{\mu}_i) = \mathbb{E}_{o,a \sim \mathcal{D}} \big[ \nabla_{\theta_i} \boldsymbol{\mu}_i(a_i|o_i) \nabla_{a_i} Q_i^{\boldsymbol{\mu}}(o_1, a_1, \ldots, o_N, a_N)|_{a_i = \boldsymbol{\mu}_i(o_i)} \big], \qquad (2)$$

for $i^{\text{th}}$ agent. Its centralised $Q_i^{\boldsymbol{\mu}}$ function is updated to minimise a loss based on temporal-difference

$$\mathcal{L}(\theta_i) = \mathbb{E}_{o,a,r,o'} \big[ (Q_i^{\boldsymbol{\mu}}(o_1, a_1, \ldots, o_N, a_N) - y)^2 \big], \qquad (3)$$

where $y = r_i + \gamma Q_i^{\boldsymbol{\mu}'}(o_1', a_1', \ldots, o_N', a_N')\big|_{a_j' = \boldsymbol{\mu}_j'(o_j')}$. Here, $\boldsymbol{\mu}_i'$ is the target policy whose parameters $\theta_i'$ are periodically updated with $\theta_i$ and $\mathcal{D}$ is the experience replay buffer consisting of the tuples $(o, a, r, o')$, where each element is a set of size $N$, i.e. $o = \{o_1, \ldots, o_N\}$.

### 3.4 INTRINSICALLY MOTIVATED RL AND HIERARCHICAL-DQN

Intrinsically motivated learning has been well-studied in the RL literature (Singh et al., 2004; Schmidhuber, 1991); nevertheless, how to design a good intrinsic reward function is still an open question. Existing techniques relate to different notions of intrinsic reward. In general, an intrinsic reward can be considered an exploration bonus representing the novelty of the visited state. In other words, the intrinsic rewards encourage the agent to explore the state space while the extrinsic rewards collected from the environment provide task related feedback. For example, in the simplest setting, an intrinsic reward can be a decreasing function of state visitation counts (Bellemare et al., 2016; Ostrovski et al., 2017).

Kulkarni et al. (2016) introduce a notion of relational intrinsic rewards in order to train a two-level hierarchical-DQN model. In this model, at the top-level, the agent learns a policy $\boldsymbol{\pi}_g$ to select an intrinsic goal $g \in \mathcal{G}$, i.e. $\boldsymbol{\pi}_g = P(g|s)$. This intrinsic goal is then used at the bottom-level whereby the agent learns a policy $\boldsymbol{\pi}_a$ for its actions, i.e. $\boldsymbol{\pi}_a = P(a|s, g)$. In this setting, the top-level and the bottom-level policies are driven by the extrinsic and intrinsic rewards, respectively.

## 4 MULTI-AGENT DDPG WITH A COMMUNICATION MEDIUM

### 4.1 PROBLEM FORMULATION AND PROPOSED APPROACH

We consider partially observable Markov Games, and assume that the observations received by most agents are extremely noisy and weakly correlated to the true state, which makes learning optimal policies unfeasible. Conditioning each policy on all $N$ private observations, i.e. $a_i = \boldsymbol{\mu}_i(o_1, \ldots, o_N)$, is not helpful given that a large majority of $o_i$'s are uncorrelated to the corresponding $s$, i.e. they provide a poor representation of the current true state for the $i^{\text{th}}$ agent. To address this challenge, we let every agent's policy depend on its private observations as well as those explicitly and selectively shared by other agents. As agents cannot discriminate between relevant and noisy information on their own, the ability to decide whether to share their own observations with others must also be acquired through experience. More formally, we introduce two hierarchically arranged sets of policies, $\boldsymbol{\nu} = \{\boldsymbol{\nu}_1, \ldots, \boldsymbol{\nu}_N\}$ and $\boldsymbol{\mu} = \{\boldsymbol{\mu}_1, \ldots, \boldsymbol{\mu}_N\}$, that are coupled through a communication medium $m = \{m_1, \ldots, m_N\}$, where $m_i$ denotes the information shared to the $i^{\text{th}}$ agent. The *action policies* in $\boldsymbol{\mu}$ determine the actions agents take to interact with the environment, whereas the *communication policies* in $\boldsymbol{\nu}$ control the communication actions determining the information shared in the medium. At the top-level, each agent chooses a communication action $c_i$ through its communication policy $\boldsymbol{\nu}_i$ conditioned only on its own private observation, i.e. $c_i = \boldsymbol{\nu}_i(o_i)$.

We consider two possible types of communication mechanisms: *broadcasting* (one-to-all) and *unicasting* (one-to-one). In the broadcasting case, each communication action is a scalar, $c_j \in$

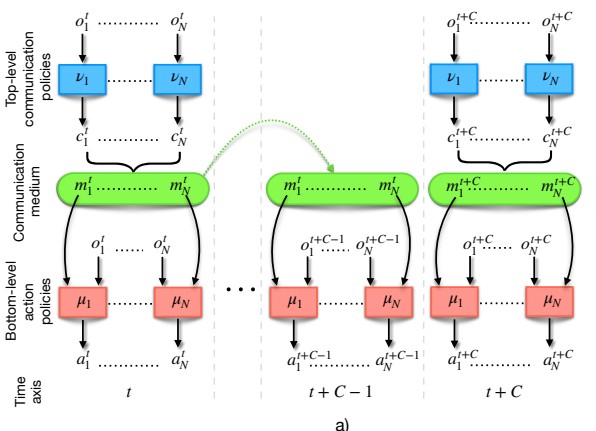 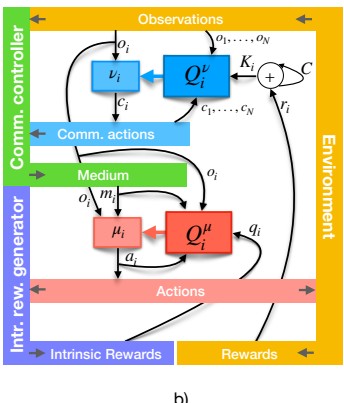

Figure 1: Overview of MADDPG-M. In (a), the $N$ agents learn two hierarchically arranged sets of policies, which are connected through a communication medium. During training, we run communication policies at a slower time-scale, i.e. once in every $C$ steps of the action policies, and determine the environmental actions using fixed communication medium over the $C$ steps. In (b), the communication policies are learned within a CTDE paradigm using the cumulative sum of the rewards collected from the environment for these $C$ steps, while we learn action policies in decentralised way using intrinsic rewards estimated with respect to the communication medium.

$\mathbb{R} \mid 0 \leq c_j \leq 1$, and the observation of the agent with the largest communication action is sent to all other agents; in this case $m$ is defined as

$$m = \left\{ m_i = o_k \quad \forall i \in \{1, \dots, N\} \quad \text{where } k = \arg\max_j (c_1, \dots, c_j, \dots, c_N) \right\} \qquad (4)$$

In the unicasting case, the communication action is an $N$-dimensional vector, i.e. $c_{j,:} \in \mathbb{R}^{1 \times N} \mid 0 \leq c_{j,i} \leq 1$ where $c_{j,i}$ can be interpreted as a measure of the $j^{\text{th}}$ agent's "willingness" to share its private observation with the $i^{\text{th}}$ agent such that the observation of the agent with the greatest willingness is shared with $i^{\text{th}}$ agent, i.e. assigned to $m_i$. In this case, $m$ is defined as

$$m = \left\{ m_i = o_k \quad \text{where } k = \arg\max_j (c_{1,i}, \dots, c_{j,i}, \dots, c_{N,i}) \right\} \qquad (5)$$

At the bottom-level, exploiting the information that has been shared, each agent determines its environmental action, i.e. $a_i = \boldsymbol{\mu}_i(o_i, m_i)$.

## 4.2 Learning Algorithm

The two sets of policies, $\boldsymbol{\nu}$ and $\boldsymbol{\mu}$, are coupled and must be learned concurrently. To address this issue, we use two different levels of temporal abstraction to collect transitions from the environment during training; that is, $\boldsymbol{\nu}$ and $\boldsymbol{\mu}$ are run at different time-scales. The communication actions are performed once in every $C$ steps. During this period, the state of medium is kept fixed, i.e $m^t = m^{t+1} = \dots = m^{t+C-1}$, and the environmental actions $a$ are obtained using the fixed medium. Given that the environment does not explicitly reward good communication strategies, there is no obvious way to optimise the communication policies. Instead, we use the cumulative sum of the extrinsic rewards collected from the environment for these $C$ steps, i.e. $K = \sum_{t'=t}^{t+C} r^{t'}$. At each time step, we also generate an *intrinsic reward*, $q$, in response to the environmental actions and use it to optimise $\boldsymbol{\mu}$.

In the RL literature, the notion of intrinsic rewards is mostly used for exploration purposes. Instead, in this work, intrinsic rewards are introduced to enable the agents to learn the environmental dynamics even when the communication decisions coming from the top-level are not optimal. In the tasks we consider, the extrinsic rewards of the environment measure the distance between the agents and their true targets even when the agents do not truly observe these targets. When the agents cannot see the true targets, they cannot learn to reach them because their observations and the received rewards become uncorrelated. On the other hand, the intrinsic rewards we introduce represent the distance between the agents and the targets that appear in the medium, regardless of whether they are the

noisy ones or the true ones. By doing this, at the bottom-level the agents learn how to reach the targets shared in the medium. At the top-level, using accumulated extrinsic rewards, they collectively infer which observations better represent the true targets and should be shared through the medium. Our developments are inspired by Kulkarni et al. (2016) where the intrinsic rewards are used to aid exploration in a single agent system. Here we introduce an analogy between their concept of intrinsic goals, which are held fixed until they are reached by the agent, and the concept of the communication medium $m$.

We keep two separate experience relay buffers, $\mathcal{D}_{\boldsymbol{\nu}}$ and $\mathcal{D}_{\boldsymbol{\mu}}$. The experience replay buffer $\mathcal{D}_{\boldsymbol{\nu}}$ consists of the tuples $(o, c, K, o'')$, where $o''$ denotes the $C^{\text{th}}$ observation after $o$, i.e. $o'' = o^{t+C}$, and provides the samples to be used to update the communication policies $\boldsymbol{\nu}$. On the other hand, the other experience replay buffer, $\mathcal{D}_{\boldsymbol{\mu}}$, consists of the tuples $(o, m, a, q, o')$ where $o'$ denotes the next observation after $o$, i.e. $o' = o^{t+1}$, and provides the samples to be used to update the action policies $\boldsymbol{\mu}$. We employ actor-critic policy gradient based methods for both $\boldsymbol{\nu}$ and $\boldsymbol{\mu}$, and train the communication policies $\boldsymbol{\nu}$ within a CTDE paradigm. For an agent, the policy gradient with respect to parameters $\theta_{\nu,i}$ is written as

$$\nabla_{\theta_{\nu,i}} J(\boldsymbol{\nu}_i) = \mathbb{E}_{o,c\sim\mathcal{D}_{\nu}}\left[\nabla_{\theta_{\nu,i}}\boldsymbol{\nu}_i(c_i|o_i)\nabla_{c_i}Q_i^{\boldsymbol{\nu}}(o_1, c_1, \ldots, o_N, c_N)|_{c_i=\boldsymbol{\nu}_i(o_i)}\right]. \tag{6}$$

The corresponding centralised action-value function $Q_i^{\boldsymbol{\nu}}$ is updated to minimise the following loss based on temporal-difference

$$\mathcal{L}(\theta_{\nu,i}) = \mathbb{E}_{o,c,K,o''}\left[(Q_i^{\boldsymbol{\nu}}(o_1, c_1, \ldots, o_N, c_N) - y)^2\right], \tag{7}$$

where $y = K_i + \gamma Q_i^{\boldsymbol{\nu}'}(o_1'', c_1'', \ldots, o_N'', c_N'')\big|_{c_j''=\boldsymbol{\nu}_j'(o_j'')}$ and $\boldsymbol{\nu}_i'$ is the target policy whose parameters $\theta_{\boldsymbol{\nu},i}'$ are periodically updated with $\theta_{\boldsymbol{\nu},i}$. The action policies $\boldsymbol{\mu}$ at the bottom-level generalise not only over the private observations, but also over $m$. The policy gradient with respect to parameters $\theta_{\mu,i}$ then becomes

$$\nabla_{\theta_{\mu,i}} J(\boldsymbol{\mu}_i) = \mathbb{E}_{o,m,a\sim\mathcal{D}_{\mu}}\left[\nabla_{\theta_{\mu,i}}\boldsymbol{\mu}_i(a_i|o_i, m_i)\nabla_{a_i}Q_i^{\boldsymbol{\mu}}(o_i, m_i, a_i)|_{a_i=\boldsymbol{\mu}_i(o_i,m_i)}\right], \tag{8}$$

Unlike the communication policies, the action policies are trained in a decentralised manner. The corresponding action-value function $Q_i^{\boldsymbol{\mu}}$ of the $i^{\text{th}}$ agent is updated to minimise the following loss based on temporal-differences

$$\mathcal{L}(\theta_{\mu,i}) = \mathbb{E}_{o,m,a,q,o'}\left[(Q_i^{\boldsymbol{\mu}}(o_i, m_i, a_i) - y)^2\right], \tag{9}$$

where $y = q_i + \gamma Q_i^{\boldsymbol{\mu}'}(o_i', m_i, a_i')\big|_{a_i'=\boldsymbol{\mu}_i'(o_i',m_i)}$ and $\boldsymbol{\mu}_i'$ is the target policy whose parameters $\theta_{\boldsymbol{\mu},i}'$ are periodically updated with $\theta_{\boldsymbol{\mu},i}$. An illustration of the proposed approach can be found in Figure 1, and the pseudo-code describing the learning algorithm can be found in the Appendix.

## 5 EXPERIMENTS

### 5.1 ENVIRONMENTS

In this section we introduce six different variations of the *Cooperative Navigation* problem from the Multi-agent Particle Environment (Lowe et al., 2017). In its original version, $N$ agents need to learn to reach $N$ landmarks while avoiding collisions with each other. Each agent observes the relative positions of the other $N-1$ agents and the $N$ landmarks. The agents are collectively rewarded based on their distance to the landmarks. Unlike most real-world multi-agent use cases, each agent's private observation provides an almost complete representation of the true state of the environment. As such, independently trained agents can reach performance levels comparable to those achievable through centralised learning (see the Appendix for supporting evidence). Hence, in its original version, there is no real need for inter-agent communication. We now describe our modifications of this environment that more closely capture the complexities of real-world applications. We classify the scenarios into two groups according to the type of the communication strategy required to solve the task.

In the first group, only one of the agents - the *gifted agent* - can observe the true position of the landmarks. This special agent can either remain the same throughout the whole learning period or vary across episodes, and even within an episode. All other agents besides the gifted one receive inaccurate information about the landmarks' positions. Crucially, no agent is aware of their status (i.e. whether

they are gifted or not), rather they all need to learn through interactions with the environment whether their own observations truly contribute to the improvement of the overall policies, and should therefore be shared with others. This learning task is accomplished by deciding at any given time whether to pass their observations onto all other agents simultaneously through a broadcasting communication mechanism; see also Eq. (4). Only indirect feedback from the environment through the reward function can inform the agents as to whether their current communication strategy improves their policies. This group of tasks include three different variants of increasing complexity depending on how the gifted agent is defined: in the *fixed* case, the gifted agent stays the same throughout the training phase, i.e. the true landmarks are always observed by the same agent; in the *alternating* case, the gifted agent may change at each episode, i.e. the ability to observe the true landmarks is randomly assigned to one of the agents at the beginning of each episode and represented by a *flag* in their observation space; in the *dynamic* case, the agent closest to the centre of the environment becomes the gifted one within each episode. Through the relative distances between each other, agents need to understand implicitly which one of them is closest to the centre.

The second group of environments is characterised by the fact that each landmark has been pre-assigned to a particular agent, and an agent needs to occupy its allocated landmark to collect good rewards. In these scenarios, each agent correctly observes only the location of one landmark and the other landmarks are wrongly perceived. The agents are again unaware of their true status (i.e. they do not know which one of the landmarks is true, and dedicated to whom), and must learn through experience how to strategically share information so as to maximise the expected rewards. In these settings an agent can decide to send its observation, at any given time, to which one(s) of the $N-1$ remaining agents through a unicasting mechanism; see also Eq. (5). Within this group we also have three different variants of increasing complexity depending on how frequently the correct observation dependencies change; *fixed* throughout the training, *alternating* across episodes and *dynamic* within each episode according to the agents' distances to the centre of the environment. In all 6 scenarios, while the extrinsic rewards of the environment represent the distance to the true landmark locations to be occupied, the intrinsic rewards we generate represent the distance to the landmark locations shared in the medium, regardless of whether or not they are actually the true landmarks.

## 5.2 COMPARISON WITH BASELINES

We evaluate MADDPG-M against four actor-critic based baselines - DDPG, MADDPG, Meta-agent and DDPG-OC - on the six environments introduced in the previous section. In all experiments we use three agents, i.e. $N = 3$. In DDPG, agents are trained and executed in decentralised manner, i.e. $Q_i^{\boldsymbol{\mu}}(o_i, a_i)$ and $\boldsymbol{\mu}_i(o_i)$. In MADDPG, agents are trained in centralised manner, but the actions are executed in decentralised manner as each agent's policy is conditioned only on its own observations, i.e. $Q_i^{\boldsymbol{\mu}}(o_1, a_1, \ldots, o_N, a_N)$ and $\boldsymbol{\mu}_i(o_i)$. A Meta-agent has access to all the observations, across all agents, during both training and execution, i.e. $Q_i^{\boldsymbol{\mu}}(o_1, a_1, \ldots, o_N, a_N)$ and $\boldsymbol{\mu}_i(o_1, \ldots, o_N)$. Although this approach becomes impractical with a large number of agents, it is included here to demonstrate the performance gains that can be achieved by strategically communicating only the relevant information compared to a naive solution where all the observations are shared, including the noisy ones. The DDPG-OC (DDPG with Optimal Communication) baseline is related to DDPG, but uses a hard-coded communication pattern, i.e. $m$ is assigned optimally using *a priori* knowledge about the underlying communication requirement. The agents are trained and executed in a decentralised

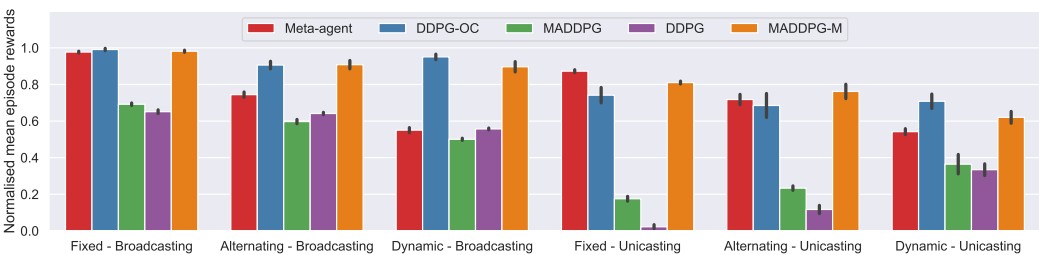

Figure 2: Comparison between MADDPG-M and 4 baselines for all 6 scenarios. Each bar cluster shows the 0-1 normalised mean episode rewards when trained using the corresponding approach, where a higher score is better for the agent. Full results are given in the Appendix.

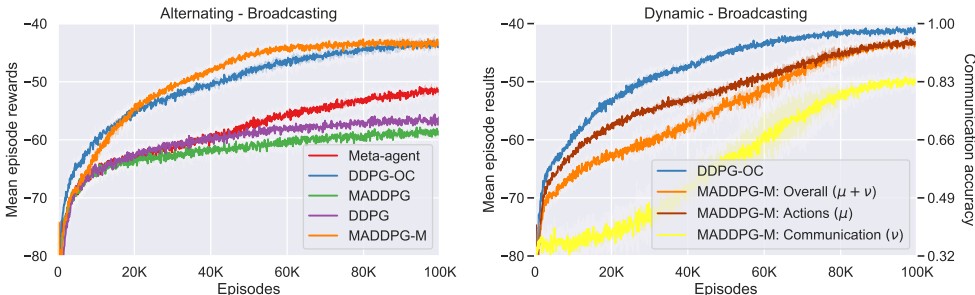

Figure 3: a) On *Alternating - Broadcasting*, the reward of MADDPG-M against baseline approaches after 100,000 episodes. b) On *Dynamic - Broadcasting*, the rewards of MADDPG-M against DDPG-OC, the accuracy curve of the communication actions to observe the individual performance of $\nu$, and the collected intrinsic rewards to observe the individual performance of $\mu$.

manner, but exploiting the communication, i.e. $Q_i^{\mu}(o_i, m_i, a_i)$ and $\mu_i(o_i, m_i)$. As the state of the medium is hard-coded, only the action policies need to be learned in this case. This approach is included to study what level of performance is achievable when communicating optimally, and learning only the main policies. Following Lowe et al. (2017), we use a two-layer ReLU Multi Layer Perceptron (MLP) with 64 units per layer to parameterise the policies, and a similar approximator using 128 units per layer to parameterise their action-value functions. We train them until they all convergence. After training, we run an additional 1,000 episodes to collect performance metrics: *collected rewards* for all baselines, in addition to *collected intrinsic rewards* and *communication accuracies* only for MADDPG-M. We repeat this process five times per baseline, and per scenario, and report the averages. Figure 2 summarises our empirical finding in terms of normalised mean episode rewards (the higher the better). Additional numerical details are provided in Appendix.

Initially, we examine the first group of scenarios consisting of tasks that can be solved using a *broadcasting* communication mechanism. Figure 3-a shows learning curves for MADDPG-M and all baselines on the *alternating-broadcasting* scenario in terms of rewards collected from the environment. In these cases, both DDPG and MADDPG fail to learn the correct behaviour; this was an expected outcome given that both methods do not allow for the observations to be shared. Their poor performances reinforce the idea that learning a coordinated behaviour through centralised training may not be sufficient in certain situations. These levels of performance provide a lower bound in our experiments. In practice, when using these baselines, we observed that the agents simply move towards the middle of the environment; even the gifted agent cannot learn a rational behaviour as the reward signal becomes noisy due to arbitrary actions taken by the agents. On the other hand, the performance achieved by DDPG-OC demonstrate that, when $m$ is correctly controlled, all the scenarios can be accomplished even when the agents are trained in a decentralised manner. Interestingly, despite reaching a satisfactory level on the simplest *fixed* case, the performance of the Meta-agent decreases dramatically as the complexity of our environments increases, and this algorithm completely fails to solve the *dynamic* case.

Conversely, MADDPG-M allows the agents to simultaneously learn the underlying communication scheme as well as the optimal action policies, and ultimately perform quite similarly to DDPG-OC in all our environments. In order to assess the action-specific (due to $\mu$) and communication-specific (due to $\nu$) performances, Figure 3-b presents the collected intrinsic rewards as well as the accuracy of the communication actions performed by MADDPG-M with respect to those optimally implemented in DDPG-OC on *dynamic-broadcasting* scenario. In the initial phases of training, although the communication policy is not yet sufficiently optimised, the MADDPG-M agents are nevertheless able to begin learning the environment dynamics and the expected actions through the intrinsic rewards. Improved environmental actions subsequently provide better feedback yielding improved communications actions, and so on. Ultimately, MADDPG-M agents perform comparably to DDPG-OC. Again, the communication accuracy decreases as the environments become more difficult. However, even in the most complex setting amongst the *broadcasting* scenarios, MADDPG-M agents choose the optimal communication actions $88.82\%$ of time, which is sufficient to accomplish the task. Very similar conclusions can be drawn when studying the *unicasting* scenarios. Due to the increased complexity, agents across all baselines tend to collect less rewards than their counterparts

in the *broadcasting* scenarios. MADDPG-M can achieve a performance similar to the empirical upper-bound provided by DDPG-OC. It is worth noting that the observed variability in the *unicasting* scenarios is higher compared to the *broadcasting* scenarios due to the increased communication requirements as well as the task complexity (e.g. each agent needs to move to the opposite side of the environment, which results in more collisions). This may explain why DDPG-OC has higher variance despite using optimal communication. Interestingly, in *dynamic-unicasting* scenario, MADDPG-M agents can only find the overall optimal communication pattern $28.98\%$ of time. However, as the individual communication actions are accurate for $64.23\%$ of time, they can manage to accomplish the task. Further results as well as implementation details (including hyperparameters) can be found in the Appendix.

## 6 Conclusions

In this paper we have studied a multi-agent reinforcement learning problem characterised by partial and extremely noisy observations. We have considered two instances of this problem: a situation where most agents receive wrong observations, and a situation where the observations required to characterise the environment are randomly distributed across agents. In both cases, we demonstrate that learning what and when to communicate is an essential skill that needs to be acquired in order to develop a collaborative behaviour and accomplish the underlying task. Our proposed MADDPG-M algorithm enables concurrent learning of an optimal communication policy and the underlying task. Effectively, the agents learn an information sharing strategy that progressively increases the collective rewards. The key technical contribution consists of hierarchical interpretation of the communication-action dependency. Agents learn two policies that are connected through a communication medium. To train these policies concurrently, we use different levels of temporal abstraction and also exploit intrinsically generated rewards according to the state of the medium. In our studies, we have considered scenarios where sharing a single observation at a time is sufficient to accomplish the task. There might be more complex cases where an agent needs to reach the observations of multiple agents at the same time. Moreover, rather than sharing raw observations, which may be high-dimensional and possibly contain redundant information (e.g. pixel data), it may be conceivable to learn a more useful representation.

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

# A APPENDICES

## A.1 MADDPG-M PSEUDO CODE

For completeness, we provide the pseudo-code for MADDPG-M in Algorithm 1.

---

**Algorithm 1:** Learning algorithm for MADDPG-M

---

Initialise parameters of $\boldsymbol{\mu}$, $\boldsymbol{\nu}$ and $Q^{\boldsymbol{\mu}}$, $Q^{\boldsymbol{\nu}}$
Initialise replay buffers $\mathcal{D}_{\boldsymbol{\mu}}$ and $\mathcal{D}_{\boldsymbol{\nu}}$
Initialise random processes $\mathcal{N}_{\boldsymbol{\mu}}$ and $\mathcal{N}_{\boldsymbol{\nu}}$ for exploration
**for** $episode \leftarrow 1$ **to** $num\_episodes$ **do**
  Reset the environment and receive initial observations $o = \{o_1, \dots, o_N\}$
  Reset temporal abstraction counter $count \leftarrow 0$
  **for** $t \leftarrow 1$ **to** $num\_steps$ **do**
    **if** $count$ is $0$ **then**
      **for** $i \leftarrow 1$ **to** $N$ **do**
        Select comm. action $c_i = \boldsymbol{\nu}_i(o_i) + \mathcal{N}_{\boldsymbol{\nu}}^t$ w.r.t. the current comm. policy and exploration
      Obtain the state of the medium $m = f_m(o, c)$ where $f_m$ is either Eq. (4) or Eq. (5)
      Keep observations $o_{init} \leftarrow o$
      Reset accumulated reward $K \leftarrow 0$
    **for** $i \leftarrow 1$ **to** $N$ **do**
      Select action $a_i = \boldsymbol{\nu}_i(o_i, m_i) + \mathcal{N}_{\boldsymbol{\mu}}^t$ w.r.t. the current action policy and exploration
    Execute actions $a = (a_1, \dots, a_N)$, get rewards $r = (r_1, \dots, r_N)$ and next obs. $o' = (o'_1, \dots, o'_N)$
    Accumulate rewards $K \leftarrow K + r$
    Get intrinsic rewards $q = f_q(o, a, o', m)$, where $q = (q_1, \dots, q_N)$
    Store $(o, m, a, q, o')$ in replay buffer $\mathcal{D}_{\boldsymbol{\mu}}$
    **if** $count$ is $C - 1$ **then**
      Store $(o_{init}, c, K, o')$ in replay buffer $\mathcal{D}_{\boldsymbol{\nu}}$
      Reset temporal abstraction counter $count \leftarrow 0$
    **else**
      Increment temporal abstraction counter $count \leftarrow count + 1$
    $o \leftarrow o'$
    **for** $i \leftarrow 1$ **to** $N$ **do**
      Sample a random mini-batch of $\mathcal{S}$ samples $(o^k, m^k, a^k, q^k, o'^k)$ from $\mathcal{D}_{\boldsymbol{\mu}}$
      Set $y^k = q_i^k + \gamma Q_i^{\boldsymbol{\mu}'}(o_i'^k, m_i^k, a_i')\big|_{a_i' = \boldsymbol{\mu}_i'(o_i'^k, m_i^k)}$
      Update action critic by minimising the loss:
      $\mathcal{L}(\theta_{\mu,i}) = \frac{1}{S} \sum_k (Q_i^{\boldsymbol{\mu}}(o_i^k, m_i^k, a_i^k) - y^k)^2$
      Update action actor using the sampled policy gradient:
      $\nabla_{\theta_{\mu,i}} J(\boldsymbol{\mu}_i) \approx \frac{1}{S} \sum_k \nabla_{\theta_{\mu,i}} \boldsymbol{\mu}_i(o_i^k, m_i^k) \nabla_{a_i} Q_i^{\boldsymbol{\mu}}(o_i^k, m_i^k, a_i)\big|_{a_i = \boldsymbol{\mu}_i(o_i^k, m_i^k)}$
      Sample a random mini-batch of $\mathcal{S}$ samples $(o^k, c^k, K^k, o''^k)$ from $\mathcal{D}_{\boldsymbol{\nu}}$
      Set $y^k = K_i^k + \gamma Q_i^{\boldsymbol{\nu}'}(o_1''^k, c_1'', \dots, o_N''^k, c_N'')\big|_{c_j'' = \boldsymbol{\nu}_j'(o_j''^k)}$
      Update communication critic by minimising the loss:
      $\mathcal{L}(\theta_{\nu,i}) = \frac{1}{S} \sum_k (Q_i^{\boldsymbol{\nu}}(o_1^k, c_1^k, \dots, o_N^k, c_N^k) - y^k)^2$
      Update communication actor using the sampled policy gradient:
      $\nabla_{\theta_{\nu,i}} J(\boldsymbol{\nu}_i) \approx \frac{1}{S} \sum_k \nabla_{\theta_{\nu,i}} \boldsymbol{\nu}_i(o_i^k) \nabla_{c_i} Q_i^{\boldsymbol{\nu}}(o_1^k, c_1^k, \dots, o_i^k, c_i \dots o_N^k, c_N^k)\big|_{c_i = \boldsymbol{\nu}_i(o_i^k)}$
    Update target network parameters for each agent $i$:
    $\theta_{\nu,i}' \leftarrow \tau \theta_{\nu,i} + (1 - \tau) \theta_{\nu,i}'$
    $\theta_{\mu,i}' \leftarrow \tau \theta_{\mu,i} + (1 - \tau) \theta_{\mu,i}'$

---

## A.2 FURTHER EXPERIMENTAL DETAILS

In all of our experiments, we use the Adam optimiser with a learning rate of 0.01 and $\tau = 0.01$ for updating the target networks. The size of the replay buffer is $10^6$ and we update the network parameters after every 100 samples added to the replay buffer. We use a batch size of 1024. During training we set $C = 5$, but to get performance metrics for execution we set it back to $C = 1$. For the exploration noise, following (Lillicrap et al., 2015), we use an Ornstein-Uhlenbeck process (Uhlenbeck & Ornstein, 1930) with $\theta = 0.15$ and $\sigma = 0.2$. For all environments, we run the

algorithms for 100,000 episodes with 25 steps each. We only change $\gamma$ across different scenarios. We use $\gamma = 0.8$ for MADDPG-M in *fixed-broadcasting*, *alternating-unicasting* and *dynamic-unicasting* scenarios. We all use $\gamma = 0.85$ for all models in all scenarios, except these three cases. The learning curves for all model in all scenarios are provided in Figure A.4 and Figure A.5. The learning curves of three baseline models, Meta-agent, MADDPG, DDPG, in the original *Cooperative Navigation* scenario of Multi-agent Particle Environments are given in Figure A.6 in order to show that there is no real need for inter-agent communication in the original *Cooperative Navigation* scenario as DDPG performs similarly to MADDPG and Meta-agent.

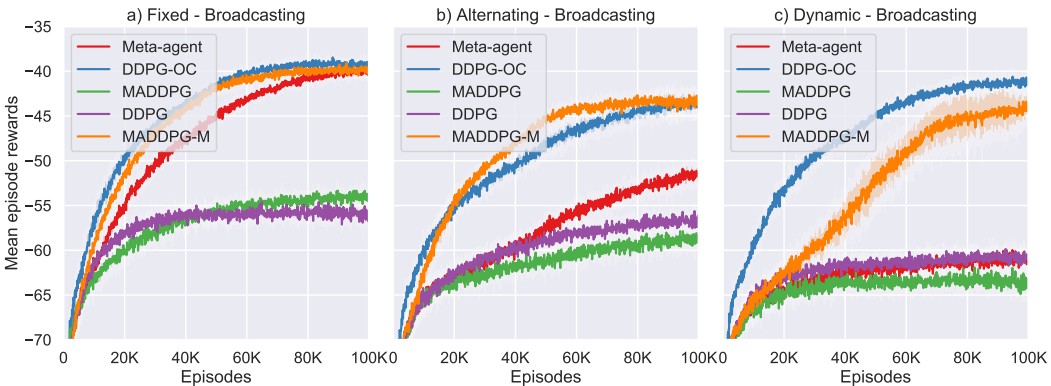

Figure A.4: Learning curves for all model in *broadcasting* scenarios over 100,000 episodes. Results of baseline models are also provided for comparison.

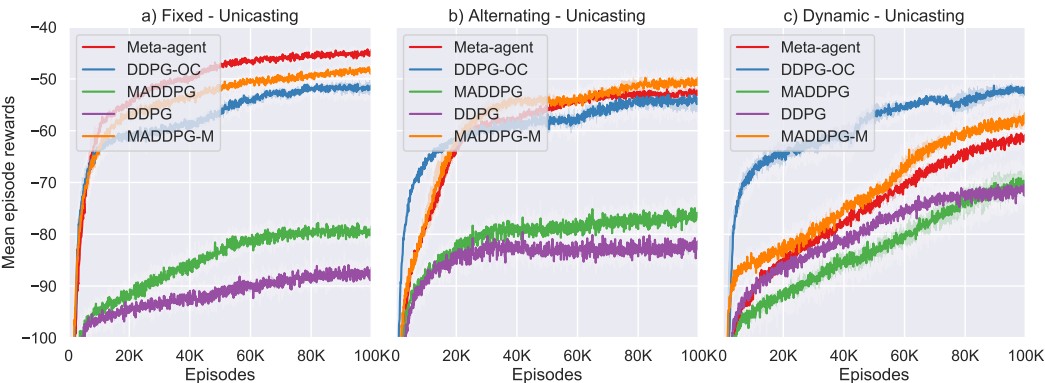

Figure A.5: Learning curves for all model in *unicasting* scenarios over 100,000 episodes. Results of baseline models are also provided for comparison.

Table A.1: Mean (standard deviations) episode rewards for all baselines in all 6 scenarios.

|  | Fixed - Broadcasting | Alternating - Broadcasting | Dynamic - Broadcasting | Fixed - Unicasting | Alternating - Unicasting | Dynamic - Unicasting |
|---|---|---|---|---|---|---|
| Meta-agent | $-39.95(\pm 4.50)$ | $-51.42(\pm 7.70)$ | $-60.98(\pm 8.82)$ | $-45.09(\pm 6.58)$ | $-52.73(\pm 6.73)$ | $-61.39(\pm 12.22)$ |
| Hard-coded | $-39.26(\pm 4.45)$ | $-43.44(\pm 5.92)$ | $-41.25(\pm 5.24)$ | $-51.57(\pm 7.03)$ | $-54.34(\pm 10.61)$ | $-53.20(\pm 8.54)$ |
| MADDPG | $-54.00(\pm 7.43)$ | $-58.67(\pm 8.90)$ | $-63.44(\pm 9.88)$ | $-79.47(\pm 12.99)$ | $-76.60(\pm 17.72)$ | $-70.15(\pm 14.07)$ |
| DDPG | $-56.00(\pm 8.96)$ | $-56.50(\pm 8.51)$ | $-60.66(\pm 8.68)$ | $-87.02(\pm 14.21)$ | $-82.35(\pm 19.50)$ | $-71.63(\pm 14.41)$ |
| Our approach | $-39.73(\pm 5.09)$ | $-43.34(\pm 7.29)$ | $-43.91(\pm 7.75)$ | $-48.16(\pm 7.02)$ | $-50.55(\pm 8.50)$ | $-57.53(\pm 8.50)$ |

### A.2.1 EFFECTS OF HYPERPARAMETERS

Figure A.7-a illustrates the learning curves for MADDPG-M with different $C$ values in *dynamic-broadcasting* scenario. All curves are obtained using $\gamma = 0.7$. One can observe that choosing $C > 2$ improves the training; however, further changes do not affect the performance significantly.

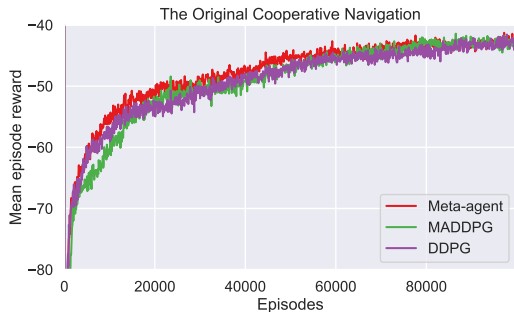

Figure A.6: Learning curves for Meta-agent, MADDPG, DDPG in the original *Cooperative Navigation* scenario over 100,000 episodes. Please note that DDPG performs similarly to MADDPG and Meta-agent.

Table A.2: Mean (standard deviation) communication accuracies for MADDPG-M in all 6 scenarios. *All* presents the overall accuracy (if all $N$ dimensions of the medium is set correctly) and *any* presents the individual accuracy (if any one of $N$ dimensions of the medium is set correctly). These two values are the same in the *broadcasting* scenarios.

|  | Fixed - Broadcasting | Alternating - Broadcasting | Dynamic - Broadcasting | Fixed - Unicasting | Alternating - Unicasting | Dynamic - Unicasting |
|---|---|---|---|---|---|---|
| All | 99.98%(±0.02) | 99.54%(±0.56) | 88.82%(±5.91) | 96.89%(±2.79) | 47.43%(±0.66) | 28.98%(±5.82) |
| Any | - | - | - | 96.96%(±0.08) | 49.30%(±1.43) | 64.23%(±2.56) |

### A.2.2 EFFECTS OF FIXING THE MEDIUM DURING TRAINING

We make an analogy between the concept of intrinsic goals introduced by Kulkarni et al. (2016), which are held fixed until they are reached by the agent, and the concept of the communication medium $m$ in this work. Nevertheless, one might also make an analogy between these intrinsic goals and the communication actions $c$, and propose to fix only $c$ while updating $m$ according to these fixed communication actions. However in this case, the state of the medium would change at each time-step with the environmental actions of the agents that are granted to change it. Therefore, this would introduce a non-stationary reference for other agents as they condition their actions on the medium, and would contribute to the overall non-stationarity of the multi-agent setting. Figure A.7-b empirically shows that fixing the medium along with the communication actions help the learning algorithm find a better policy. During training, fixing the medium for $C$ steps is crucial for the performance as it provides a stationary reference to the agents to learn their action policies conditioning on the medium. However, it is worth noting that updating the medium in a slower time-scale during training does not cause a sparse communication during execution. Instead, as we set $C = 1$ after the training, agents can communicate at every time step during execution.

### A.2.3 EFFECTS OF USING DISCRETE COMMUNICATION ACTIONS

Communication actions we consider in this paper can also be implemented as discrete actions through a Gumbel-Softmax estimator (Jang et al., 2016). However, we have observed similar performances with both cases, i.e. -43.91 with continuous communication actions vs. -44.54 with discrete communication actions, and decided to keep the continuous actions as they appear more generalisable and potentially amenable to further extensions.

### A.3 ENVIRONMENT DETAILS

Figure A.8 illustrates the scenarios considered in this paper. The details of the experimental results are shown in Tables A.1 and A.2. In all environments, each agent observes its own position and velocity, and also observes the true relative positions of other agents. The observation schemes of the landmark locations vary across the scenarios. Environmental actions of agents consist of a vector of $4$ continuous $0 - 1$ valued scalars. Each scalar corresponds to a direction and its magnitude determines the velocity of the agent in that direction.

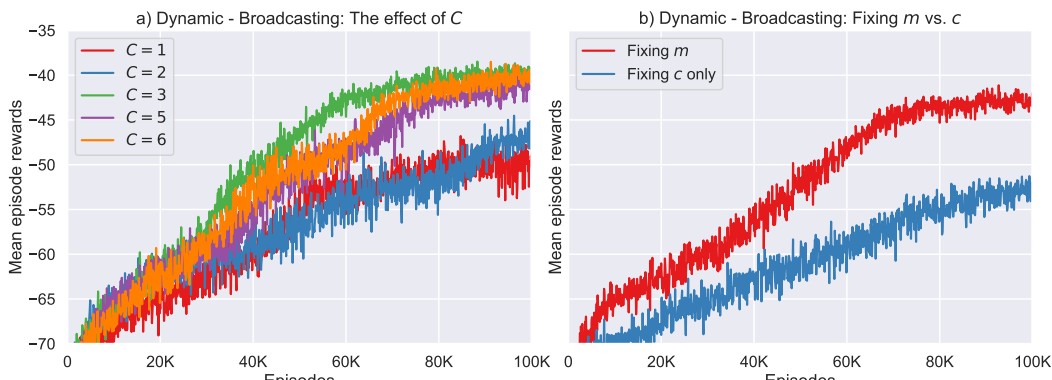

Figure A.7: a) Learning curves for MADDPG-M with different $C$ values in *dynamic-broadcasting* scenario. b) Learning curves for MADDPG-M i) when $m$ is fixed for $C$ steps ii) when $c$ is fixed and $m$ is being updated according to the fixed $c$ for $C$ steps.

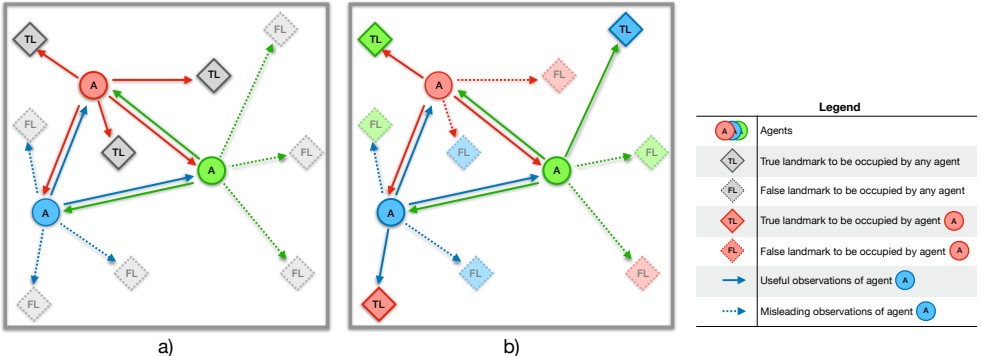

Figure A.8: An illustration of our environments: a) Within the first group, the *gifted* agent (red circle) is able to correctly observe all three landmarks (grey squares); the other agents (blue and green circles) receive the wrong landmarks' locations. b) Within the second group, each landmark is designated to a particular agent, and the agents get rewarded only when reaching their allocated landmarks. The *gift* is equally but partially distributed across the agents, and hence each agent can only correctly observe one of the landmarks (either its own or another agent's), but otherwise receives the wrong whereabouts of the remaining ones. In this case: the red agent can see the landmark assigned to the green agent, the green agent can see the landmark assigned to the blue agent, and the blue agent can see the landmark assigned to the red agent.

