# OpenReview forum: "Multi-agent Deep Reinforcement Learning with Extremely Noisy Observations"
_ICLR.cc/2019/Conference_

### Official Review · AnonReviewer2 · 2018-11-02
**Their proposed method extends MADDPG with a communication network. Concerned about unnecessary complexity of the algorithm and formalism.**

**Rating:** 6
**Confidence:** 3

**Review:**

This paper is clearly written and explains everything in a good detail. I have a few questions about the design of the algorithm and experiments that I will explain next. Most importantly, I am confused why the communication actions are modeled with continuous actions. Also, the communicating agent idea is incorporated in MADDPG paper, and the contribution of the proposed network is unclear to me.  Right now, I am leaning toward weak reject now but I might update my evaluations after seeing the authors' feedback.

1) First, your construction of communication medium simply seems to be learning a method for graph sparsification and this deserves some explanation.  Also, I think that using the graph terminology for describing the communication medium structure will significantly improve the clarity of the paper. For example, I assume that by saying that m^t = ...=m^{t+C-1} you mean that you simply fix the communication graph structure for C steps, not the communicating observations. Based on your notations, it is a little bit confusing -- in your notations $m^t$ is the set of observation that flows through the graph which should be different than $m^{t+1}$.

2) Even MADDPG is very challenging to train! Now, this paper utilizes two MADDPG, and that is something that concerns me a lot. I don't think that replicating the results of this paper is possible by other people. How much was the cost of the hyper-parameters search?

3) Why the decision of where to send the observations is modeled with a continuous control action? It can be simply modeled with discrete action in a more efficient way, right? What I mean is that $c_i$ can be a binary which tells whether send an observation or not. Am I missing anything?

4) In section 2, you argue that in the original MADDPG paper, there is no inter-agent communication. As far as I remember, they have some experiments for cooperative communication or covert communication in which the communication is allowed between the agents. I would like to know more about this statement; maybe I am missing something. Why you are not designing the communication network which is a differentiable medium such as Foerster 2015? Isn't that efficient?

5) In alternating case, I don't see (intuitively) why the communication should help to improve upon MADDPG. My intuition is that each agent will be the gifted one 1/3 on average. This means that the agents cannot perceive who gives the correct information and the policy should converge to a point where the communication does not give any new information.

6) I would like to see what will happen with C=1? I think this hyper-parameter deserves some analysis to see how it affects the performance of the proposed method.

7) In section 5, you say that in original DDPG, there is no real need for inter-agent communication". This is a little bit strict statement, I guess. For example in the case with full observability, the agents can send messages which conveys their intention and help each other.



Minor:
* I would suggest using partially observability terminology instead of saying noisy observation because I think it includes a more general class of the problems to solve.
* "that a coupled through a communication medium" -> "that are coupled through a communication medium"
* In section 4.2, it is unclear to me what is the exploration strategy. Please explain more.
* section 5.2: Using the term lower bound is not accurate. Try changing it to something else or use with quotations: "lower bound"
* What will happen you choose the top-k rule for sending the information? For example, does top-2 (two-to-one) rule improve the results? The experiments might be added in future.
-----------------------------------
post rebuttal: the authors have responded to my main questions, and I would like to increase my score, but I cannot agree with them on possile future extensions of this work, e.g. in learning from pixels.

---

> ### Author Response · Authors · 2018-11-26
> **Author feedback - Cont'd**
>
> 5) Unlike the Fixed and Dynamic cases, in the Alternating case the agents observe a flag bit showing their gift. Hence the agents need to learn to interpret their flags and to use the medium if they have the gift. It is worth noting that no such flag is needed in Dynamic case as the gifts depend on the agents’ proximity to the centre. As the reviewer anticipated, there would be no way for the agents to learn randomly assigned communication pattern, if agents didn’t have enough observation to learn the underlying rule. This is indeed very good question and shows the reviewer’s rigorous evaluation. We, again, would like to thank the reviewer for their valuable feedback. We have added this detail in the main text.
>
>  6) Choosing C > 1 helps stabilising the training. Otherwise, agents cannot understand whether the observed reward is due to the environmental actions or due to the communication decision. Briefly, we have observed that the agents are not able to learn optimal behaviours when C=1. In the revised version, we have provided the analysis showing how the performance is affected by C in the Appendices. It is also worth noting here that setting C>1 is only a strategy to help the training process. During execution, the agents update their communication actions (i.e. the medium is updated) at every time step, using C=1.
>
> 7) We believe there might be some misunderstanding related to this statement. We use this statement in Section 5.1 where we describe the environments, “Hence, in its original version, there is no real need for inter-agent communication.” However, we use this statement for the original Cooperative Navigation problem considered in [2], not for DDPG. Further evidence supporting this statement can be found in the Appendices.
>
> Minor:
> * As we denoted in the Background, our setting is Partially Observable Markov Games. However, by using ‘noisy’ term, we wanted to emphasise that there also exists an additional challenge in our setting beyond POMG
> * In section 4.2, it is unclear to me what is the exploration strategy. Please explain more.
> Additional explanation has been added for the followed exploration strategy in the Appendices.
> * As explained above, using top-k rule for sending the information is one of our possible extension ideas in future works. It might be worth noting again that these types of extension ideas motivate us to use continuous communication actions instead of discrete ones.
>
> [1] Tejas D. Kulkarni, Karthik Narasimhan, Ardavan Saeedi, and Josh Tenenbaum. Hierarchical deep reinforcement learning: Integrating temporal abstraction and intrinsic motivation.
> [2] Ryan Lowe, Yi Wu, Aviv Tamar, Jean Harb, Pieter Abbeel, and Igor Mordatch. Multi-agent actor-critic for mixed cooperative-competitive environments.
> [3] E. Jang, S. Gu, and B. Poole. Categorical reparameterization with gumbel-softmax.

---

> ### Author Response · Authors · 2018-11-26
> **Author feedback**
>
> We thank the reviewer for their comments and valuable feedback. We address the issues raised by the reviewer below.
>
> 1) As the reviewer suggested it can be said that the communication actions $(c_1, c_2,…,c_N)$ define a communication graph: the agents are to nodes and each communication action is represented by an edge. However, during training, we fix not only this communication graph structure, but also the communicating observations (i.e. the medium, as defined in the paper). We do not update the medium for these C steps. We explain this in the paper as “The communication actions are performed once in every C steps. During this period, the state of medium is kept fixed, i.e $m_t = m_{t+1} = . . . = m_{t+C −1}$, and the environmental actions $a$ are obtained using the fixed medium.”. We are confident that the notation as well as the pseudo code correctly describe the implemented idea.
>
> We have now added a discussion to the Appendices and provided empirical evidence showing that the other case would exacerbate the training.
>
> 2) We appreciate these comments, although our experience with MADDPG has been different and we have not experienced issues in training the algorithm by following the hyper parameters originally presented in [2]. When testing our architecture, we did not over optimise the hyper-parameters, we focused mostly on tuning the C value and the discount factor. We haven’t faced particular challenges in training the two hierarchically arranged MADDPGs either. We believe that the hierarchical approach we follow (i.e. fixing the medium for C steps, getting the actions using the fixed medium and rewarding them according to the intrinsic rewards) may help stabilise the training. We are confident that other researchers would be able to easily reproduce our results as we explain in the Appendices. We will also be making the code publicly available after the reviewing process is completed (so as not to violate anonymity). As a further information, we have provided the analysis showing how the performance is affected by C in the appendices.
>
> 3) Yes, the reviewer is right, and this is an important aspect that was not clearly stated in the paper. We decided to use continuous actions to define binary communication decisions. Our first implementation of the model did use discrete communication actions through a Gumbel-Softmax estimator [3]. We then attempted to use continuous actions and found that they yielded very similar performance. For completeness, we have now added the results obtained from a model adopting discrete communication actions and a corresponding discussion in the Appendices.
>
> In the revised version, we have decided to keep the continuous communication actions as they appear more generalisable and potentially amenable to further extensions, e.g. the medium could be defined as a weighed sum of the observations. We could also allow for more than one observation to be shared over the medium at the same time (this is similar to top-k idea suggested by the reviewer).
>
> 4) In the original MADDPG paper, the communication in those scenarios is hard-wired by the environment.  MADDPG by itself does not provide any inter-agent communication. Unlike the scenarios we consider, agents do not learn a communication pattern (i.e. which agent should communicate with whom).
>
> For instance, let us consider the Cooperative Communication environment described in [2]. There are two agents: a speaker and a listener. The speaker is a non-moving agent whose action is a discrete communication message. This action directly goes into the observation space of the listener who can move inside the environment. These two agents do communicate with each other, however this is not a contribution brought by the MADDPG. Instead, this communication is explicitly hard-wired by the environment. The MADDPG agents do not use any other information except their own observations during the execution. In this respect, MADDPG by itself does not provide a mechanism for inter-agent communication.
>
> There are two reasons why we haven’t design a differentiable medium such as DIAL (Foerster et al., 2016): (i) In such a case, the number of communication paths grows quadratically with the number of agents (each agent will have (N-1) x size_of_each_message additional input), and it would be hard to think of this approach as ‘decentralised’; (ii) Foerster (et al., 2016) use environments which can be solved by 1-bit (discretised 1d) messages, however in our environments the agents need to share 14-d real valued messages.

---

### Official Review · AnonReviewer3 · 2018-11-02
**Extrinsic reward and intrinsic reward are confusing**

**Rating:** 3
**Confidence:** 4

**Review:**

This paper studies multi-agent reinforcement learning where the agents need to communicate information when observations are noisy.  The agents thus need to learn what information should be sent to other agents.

The authors claim "we do not assume the existence of explicit rewards guiding the communication action," which however is questionable.  The "extrinsic reward" used to guide the communication action is simply the cumulative reward between two communication actions.  The reward is explicitly given.

The key assumption is that communication is not performed every step.  Then standard cumulative reward until the next communication can be used as immediate reward for the previous communication.  Should this assumption be considered as an assumption of the domain where the proposed approach can be applied, or is this assumption rather a technique that one should use even when communication can be performed every step?  In the latter case, the effectiveness is sparse communication is not demonstrated.

In addition, the intrinsic reward for guiding environmental actions is unclear.  In the experiment, the standard reward is simply used as intrinsic reward.  So, intrinsic reward is just standard (extrinsic) reward?  In general, how should we design intrinsic reward?  What is the advantage of not using the standard reward as intrinsic reward?

The experimental settings are too ideal for the proposed approach, and it is unclear how the proposed approach work in practical settings.  In particular, sequential decision making is not essential in the experimental settings.  What are the real applications in mind?

---

> ### Author Response · Authors · 2018-11-26
> **Author feedback**
>
> We’d like to thank the reviewer for the feedback, and we’ve addressed each one of their comments below.
>
> Firstly, we’d like to comment on the existence of explicit rewards guiding the communication action. Our claim "we do not assume the existence of explicit rewards guiding the communication action" is not a subjective assessment, hence and is not questionable - this is an explicit assumption we make in our paper. In the scenarios we describe, the rewards gathered from the environment do not provide any direct feedback regarding the communication policy. This is contrast to other work on multi-agent RL that has been presented in the literature (e.g. [1, 2]) where a communication policy is explicitly rewarded. We realise that this distinction was not sufficiently emphasised in the paper, and have attempted to clarify this in the revised version.
>
> Secondly, the reviewer has made a comment on “sparse communication”. In this work, we do not argue for the effectiveness of a sparse communication strategy. In fact, after training the algorithm, we always set C=1 in all our experiments, as explained in the paper. Setting C>1 is only used during training to implement a strategy based on different temporal abstractions. As we mentioned in the paper, this approach has been used by [3] to aid exploration in a single agent system. We draw a parallel between the concept of a medium in our work and the concept of intrinsic goals introduced in [3], and our developments follow a similar approach. Using different temporal abstractions helps stabilise the learning process. Our aim is not to achieve sparse communication. After training, setting C=1 means that the agents can communicate at every state. To address this point, we have provided an additional study in the Appendices showing how the performance is affected by C.
>
> Thirdly, we’d like to address the difference between intrinsic and extrinsic rewards as used in our work. As pointed out above, the extrinsic rewards are used to capture the proximity to the true landmarks whereas the intrinsic rewards show the proximity to the landmarks shared in the medium. We have now added a more detailed explanation to clarify this distinction. We have also added a Background subsection (3.4) showing the large body of work on intrinsically motivated learning in RL literature.  We’d also like to emphasise that defining a good intrinsic reward/goal is an open research question for RL and a general discussion of how intrinsic reward should be designed is beyond the scope of this paper. Our proposal is limited to multi-agent communication with MDPs characterised by partial and very noisy observations. For the corresponding explanation please see the second paragraph of the Section 4.2
>
> On applications in mind:
>
> The type of setting we envisage can occur in several real-world scenarios. For instance, an autonomous driving agent might not be able to observe an accident ahead due to poor visibility stemmed from weather conditions; however, observation of another vehicle in front, might be better representing the true state and hence can help decide on optimal actions. Furthermore, learning which observations to be shared is very crucial as there may be many other vehicles with limited visibility. Other examples would include robotics applications involving several coordinating agents operating in extreme conditions (e.g. under water) where there is a high probability that their sensors may be malfunctioning. Our initial work is focused on the methodological issues, and we agree with the reviewer that the need for these methods could have been better motivated should more space been available.
>
>
>  [1] Angeliki Lazaridou, Karl Moritz Hermann, Karl Tuyls, and Stephen Clark. Emergence of linguistic communication from referential games with symbolic and pixel input. arXiv preprint arXiv:1804.03984, 2018.
> [2] Igor Mordatch and Pieter Abbeel. Emergence of grounded compositional language in multi-agent populations. In Proceedings of the Thirty-Second AAAI Conference on Artificial Intelligence, New Orleans, Louisiana, USA, February 2-7, 2018, 2018.
> [3] Tejas D. Kulkarni, Karthik Narasimhan, Ardavan Saeedi, and Josh Tenenbaum. Hierarchical deep reinforcement learning: Integrating temporal abstraction and intrinsic motivation.

---

> > ### Comment · AnonReviewer3 · 2018-12-05
> > **Explicit rewards guiding the communication action**
> >
> > In the beginning of Section 4.2, you say "Given that the environment does not explicitly reward good communication strategies, there is no obvious way to optimise the communication policies. Instead, we use the cumulative sum of the extrinsic rewards collected from the environment for these C steps".  How can you say that the cumulative sum of the extrinsic rewards are not explicit rewards?

---

> > > ### Author Response · Authors · 2018-12-05
> > > **Author feedback: Explicit rewards guiding the communication action**
> > >
> > > As a general comment, we note that the Reviewer’s assessment seems to be entirely based on our choice of terms, like intrinsic/extrinsic rewards and implicit/explicit feedback for the communication strategy, rather than a technical flaw with our approach.
> > >
> > > We have used the term ‘intrinsic reward’ because the reward is associated to the particular medium being used. Although the analytical expression is the same as the environmental rewards (i.e. Euclidean distance to landmarks, in our examples), the actual numerical values are different. In our environments, these intrinsic rewards are used by the agents to learn to move towards shared landmarks even when the communication decisions are wrong, i.e. the location of the landmarks as shared amongst agents through the medium are wrong. On the other hand, the environment provides "extrinsic" rewards capturing the proximity to the “true” landmarks.
> > >
> > > We have decided to use the explicit/implicit terminology to emphasise the difference between our work and the existing articles in the literature [1, 2]. Unlike the communication-based rewards used in other articles, no direct (explicit) feedback for the communication decisions is captured in these purely physical distance-based extrinsic rewards.
> > >
> > > However, we accept that this terminology can generate confusion and are happy to use “communication-based reward” instead of “explicit”. Then the sentence in the beginning of the section 4.2 becomes
> > >
> > > "Unlike [1] and [2], given that our environments do not provide a communication-based rewards to learn good communication strategies, there is no obvious way to optimise the communication policies. Instead, we use the cumulative sum of the extrinsic rewards collected from the environment for these C steps."
> > >
> > >  [1] Angeliki Lazaridou, Karl Moritz Hermann, Karl Tuyls, and Stephen Clark. Emergence of linguistic communication from referential games with symbolic and pixel input. arXiv preprint arXiv:1804.03984, 2018.
> > > [2] Igor Mordatch and Pieter Abbeel. Emergence of grounded compositional language in multi-agent populations. In Proceedings of the Thirty-Second AAAI Conference on Artificial Intelligence, New Orleans, Louisiana, USA, February 2-7, 2018, 2018.

---

> > ### Comment · AnonReviewer3 · 2018-12-05
> > **Re: Sparse communication**
> >
> > If the agents can share all of the information at every step, it reduces to the setting of central control.  In practice, sparse communication is desirable, because it is often intractable to share all of the information.  So, it is good that the proposed approach works well with C>1.

---

> > > ### Author Response · Authors · 2018-12-05
> > > **Author feedback: Sparse communication**
> > >
> > > This comment on ‘sparse’ communication seems to suggest that the Reviewer continues to misinterpret the meaning of the hyperparameter C, and more generally how our methodology works. We had made an attempt to clarify this point in our previous comment, and an alternative explanation is in order.
> > >
> > > During training, the agents learn a communication policy controlling which observations should be shared between them at every C time step. As we empirically show, C>1 in training helps find better policies. However, during execution, the agents can decide what is shared amongst them at every time step (i.e. C=1). Our approach results in “sparse communication” (i.e. not every observation an agent acquires is shared with all the other agents at every time step), and is indeed the main contribution of this study. We have also discussed at length the “central control” scenario (presented as the Meta-agent baseline in our Experiments section), and have presented extensive comparisons with that setting.
> > >
> > > The Reviewer is not flagging a technical flaw with our method, but is rather commenting on the interpretation of a hyper-parameter, which we have hopefully addressed in the revised version. The practical benefits of 'sparse communication' as introduced in this paper have been supported by extensive experimental evidence we have presented using several environments characterised by partial and noisy observations whereby the proposed communication strategy allows the agent to learn the task compared to other baselines.

---

### Official Review · AnonReviewer1 · 2018-11-03
**Review for Paper "Multi-agent Deep Reinforcement Learning with Extremely Noisy Observations"**

**Rating:** 7
**Confidence:** 2

**Review:**

This paper addresses the challenge of learning in extremely noisy environments. The fundamental idea is to combine deep reinforcement learning of individuals, in which individuals can choose whether they share information in order to maximise the overall reward, which is a substantial difference from existing solutions in the area. To achieve this, the authors propose a hierarchical approach in which agents learn from experience, before deciding whether to share information.

To explore the performance, the authors modify an existing scenario and implement baselines that represent idealised outcomes and contemporary approaches with varying levels of communication among agents. The proposed approach performs favourable compared to alternative approaches, despite its strongly decentralised operation, and is surprisingly close (and in some cases exceeds) the ideal solution with optimal communication.

The paper is well structured and systematic in the introduction of the underlying concepts in order to retrace the complex architectural setup. Experiment and alternative architectures are described in sufficient level of detail.

The quality of the presentation is high and accessible. Prospects for future work are highlighted. At this stage, observations are limited to a single observation at a time. The authors could be more explicit about potential further challenges in using the current solution and discuss its versatility in other scenarios. However, overall, the described hierarchical approach provides an interesting avenue to address the issue of noisy observations, which warrants discussion.

---

> ### Author Response · Authors · 2018-11-26
> **Author feedback**
>
> We’d like to thank the reviewer for their time spent on our article and their positive feedback and encouraging remarks. A future challenge for us will be to assess the performance of the proposed solution in real-world applications where partial and noisy observations are quite common. In the environments we presented here, for instance, the model has been tested under extreme conditions where most agents can’t see the true locations and learning is particularly challenging. Scenarios with a very large number of agents may also be challenging as the MDPs would suffer from higher non-stationarity.

---

### Author Response · Authors · 2018-11-26
**Revision 2 and the summary of the changes**

Minor changes (mostly additions) have been made in the Revision 2 of this article. Please also note that although the changes are minor, and the placement of some parts of the text have been changed, pdfdiff mistakenly shows some additional differences.

1.	We added an additional background section on Intrinsically Motivated RL and Hierarchical-DQN.
2.	We added an additional paragraph to better clarify the notion of intrinsic/extrinsic reward in our setting, where we also provided a better intuition on how hierarchically arranged agents can overcome the problem of noisy observations.
3.	Some minor typos have been corrected.
4.	Additional materials have been added to Appendices to discuss
     a.        the effect of the hyperparameter C
     b.	the effect of discrete communication actions
     c.	the effect of fixing c rather than m

---

### Meta-Review · Area_Chair1 · 2018-12-15
**An interesting take on partially observable MARL, without enough supporting evidence**

**Confidence:** 3
**Recommendation:** Reject

**Metareview:**

The paper presents an extension of MADDPG, adding communication between agents. The methods targets extremely noisy observations settings, so that agents need to decide if they communicate their private observations (or not). There is no intrinsic/explicit reward to guide the learning of the communication, only the extrinsic/implicit reward of the downstream task.

The paper is clear and easy to follow, in particular after the updated writing. I believe some of the reviewers' points were addressed by the rebuttal. Nonetheless, some of the weaknesses of the paper still hold: namely the complexity of the approach compounded with a very specific experimental evaluation. The more complex an approach is (and it may be justified by the complexity of the setting!), the more varied its supporting evidence should be.

In its current form, the paper would constitute a good workshop contribution (to discuss the approach), but I believe it needs more varied (and/or harder) experiments to be published at ICLR.